# Stunting Malnutrition Associated with Severe Tooth Decay in Cambodian Toddlers

**DOI:** 10.3390/nu13020290

**Published:** 2021-01-20

**Authors:** Eva Peris Renggli, Bathsheba Turton, Karen Sokal-Gutierrez, Gabriela Hondru, Tepirou Chher, Sithan Hak, Etienne Poirot, Arnaud Laillou

**Affiliations:** 1Centre for International Health Protection, Robert Koch Institute, 13353 Berlin, Germany; 2Department of Dentistry, University Puthisistra, 12211 Phnom Penh, Cambodia; bethy.turton@gmail.com; 3School of Public Health, University of California, Berkeley, CA 94720, USA; ksokalg@berkeley.edu; 4UNICEF Cambodia, 12100 Phnom Penh, Cambodia; gabriela.hondru@gmail.com; 5Oral Health Bureau, Department of Preventive Medicine, Ministry of Health, 12211 Phnom Penh, Cambodia; tepirou@yahoo.com; 6Department of Preventive Medicine, Ministry of Health, 12211 Phnom Penh, Cambodia; sithan_hak@yahoo.com; 7UNICEF Chad, 1146 N’Djamena, Chad; epoirot@unicef.org; 8UNICEF Ethiopia, 1169, Addis Ababa, Ethiopia; alaillou@unicef.org

**Keywords:** severe caries, tooth decay, dental, early childhood, early childhood caries, malnutrition, undernutrition, stunting, growth and development

## Abstract

Background: The persistently high prevalence of undernutrition in Cambodia, in particular stunting or chronic malnutrition, calls for innovative investigation into the risk factors that affect children’s growth during critical phases of development. Methods: Secondary data analysis was performed on a subgroup of children who were present at two time points within the Cambodian Health and Nutrition Monitoring Study (CAHENMS) and who were less than 24 months of age at the nominated baseline. Data consisted of parent interviews on sociodemographic characteristics and feeding practices, and clinical measures for anthropometric measures and dental status. Logistic regression modelling was used to examine the associations between severe dental caries (tooth decay)—as indicated by the Significant Caries Index—and the presence of new cases of stunting malnutrition at follow-up. Results: There were 1595 children who met the inclusion criteria and 1307 (81.9%) were followed after one year. At baseline, 14.4% of the children had severe dental caries, 25.6% presented with stunted growth. 17.6% of the children transitioned from healthy status to a low height-for-age over the observation period. Children with severe dental caries had nearly double the risk (OR = 1.8; CI 1.0–3.0) of making that transition. Conclusion: Severe caries experience was associated with poorer childhood growth and, as such, could be an underinvestigated contributor to stunting.

## 1. Introduction

Over recent decades, early childhood undernutrition has been declining globally. However, high rates of stunting have persisted, while the rates of obesity have increased [1,2,3]. The modern challenge of the “double burden” of undernutrition and obesity calls for an examination of the neglected risk factors that may contribute to the high prevalence of child malnutrition. Globalization and urbanization have led to a global nutrition transition, dramatically increasing young children’s consumption of sugary drinks and snacks, contributing to both child obesity and tooth decay (dental caries) [4]. 

Dental caries is a process which leads to the destruction of tooth tissue. Dental cavities result when net demineralization occurs at the surface of the tooth in response to pH changes in the microbial biofilm (plaque). When the biofilm is exposed to free sugars, then acid is produced, and in the absence of protective factors such as tooth brushing with fluoride toothpaste, then the tooth surface will break down and carious lesions will progress [5,6]. When a carious lesion is present among children less than 6 years of age, it is commonly referred to in the literature as early childhood caries (ECC) [7]. Globally, dental caries is the most prevalent chronic disease, affecting 60–90% of schoolchildren [8]. Among the youngest age groups, most carious lesions (dental cavities) remain untreated due to limited access to dental care, particularly among low- and middle-income populations and socioeconomically disadvantaged groups [9,10,11]. Severe and uncontrolled dental caries can lead to oral infection and inflammation (abscesses), which can cause mouth pain, decreased appetite, inability to chew food, inadequate sleep, and chronic inflammation if persistent over time [12,13,14]. These factors, added to well-known aspects such as child nutrition, breastfeeding, infectious diseases, mother’s health, and psychosocial stimulation, could all contribute to undernutrition [15]. 

In Cambodia, the current rates of both early childhood undernutrition and dental caries are among the highest in the world. Contributors include Cambodia´s long period of war and humanitarian crises followed by rapid economic growth since the late 1990s, involving the introduction of ultraprocessed foods and beverages, and a dramatic increase in childhood sugar consumption [16,17]. Among Cambodian children under age 5, 32% experience stunting and 10% wasting malnutrition [18], and the prevalence of dental caries exceeds 90% [19]. Among 3–5-year-olds, 16.1% had one or more severe, deep caries infecting the surrounding soft tissues and commonly causing mouth pain, and the prevalence of this type of dental infection increases to 86% among 6-year-old children [20,21,22]. In Cambodia, the high prevalence of severe caries experience by age 2, within the first 1000 days of life, elevates the concern about the potential for adverse effects on children’s short-term and long-term growth and development. In fact, stunting, an indicator for chronic malnutrition, has been described to peak between 1 and 2 years of age [23]. Nevertheless, recent studies have proposed that the prevalence of stunting can increase beyond 24 months of age in suboptimal environments [24,25,26,27].

Given the high rates of chronic malnutrition and dental caries among Cambodian children, there is a pressing need to explore the potential relationship between these two conditions. This could help to better understand the role of severe dental caries as an under-recognized contributor to stunting. This longitudinal study examines the relationship between severe dental caries and anthropometric changes over a one-year period, in children under 2 years of age at baseline.

## 2. Materials and Methods 

This is a secondary analysis of a longitudinal cohort study, the Cambodian Health and Nutrition Monitoring Study (CAHENMS), and an added oral health component (Figure A1). Data were collected from three Cambodian provinces: two predominantly rural northeastern provinces, Ratanakiri and Kratie; and an urban area, the capital Phnom Penh. Data were collected in 2017 (baseline) and 2018 (follow-up) using 8 trained teams specialized in conducting questionnaires, anthropometric measurements, and intraoral examinations.

The original protocol for the CAHENMS was reviewed by the National Ethics Committee for Health Research, Ministry of Health, Cambodia, prior to collecting data (117/NECHR). Written consent was obtained from the parents of the participants before the baseline data collection, and verbal consent was obtained at each subsequent contact. A data-sharing agreement was made prior to the transfer of de-identified data. This is a secondary analysis of a de-identified dataset and was considered research on nonhuman subjects. Information about the original CAHENMS study is provided in the supplement.

The study population consists of a final sub-sample of 1307 children <24 months of age at baseline and approximately one year older at follow-up, as presented in Figure 1. Children lost to attrition (*N* = 12.5%), older than 23.9 months at baseline (*N* = 30.5%) and with missing anthropometric measurements were not considered in the present analysis (*N* = 0.5%). 

G*power (version 3.9.1.2) was used to calculate implied power using Chi-squared tests given a sample size of 1307 participants and the intention to observe a clinically significant 5% difference in new cases of stunting (Height-for-age *Z*-score, HAZ < −2) from 30% down to 25%. The present sample size implies a 98.6% chance of detecting a 5% difference in the incidence of stunting.

### 2.1. Questionnaires 

The 2 sets of questionnaires for sociodemographic characteristics and child feeding practices were administered at baseline by trained Cambodian interviewers in Khmer, or translated into the local indigenous language for some minority populations in northeastern provinces. Dietary intake was assessed using a 24-h recall period, based on WHO guidelines and described in published articles on child feeding practices status [28,29,30]. Parents/caregivers were asked to recall and report on breastfeeding frequency, consumed food groups, frequency of meals, and amount of food given during meals using as unit of measure context relevant utensils: spoons and Chan Chang Koeh, a traditional bowl in Southeast Asia.

To assess socioeconomic status (SES), questions were asked about assets, employment, and household characteristics to further calculate the wealth index using Principal Component Analysis (PCA) based on the validated tool performed by Filmer and Pritchett without expenditure data [31].

### 2.2. Clinical Examinations

Separate teams performed anthropometric measurements, and intraoral examinations and data were recorded on separate devices to ensure that examiners were blinded to the participants’ status. Anthropometric measures were assessed in duplicates according to the WHO guidelines, and the mean value was further used [32]. Recumbent length was recorded in children under 2 years of age or unable to stand up, and in children above the age of 2 years height was recorded. Weight was measured with a calibrated precise scale for mother and child. 

Intraoral examinations were performed by one of 8 calibrated examiners with the help of a trained assistant. All examiners achieved a kappa score of >0.9, indicating near-perfect agreement. The examination was done in supine position with a handheld torch and mouth mirror. The decayed, missing, and filled teeth (dmft) index as defined by WHO was used to measure dental caries experience [33].

### 2.3. Data Analysis

Data were delivered through Microsoft Excel and entered into IBM SPSS 25. Statistical significance was considered for *p*-values below 0.05, and no imputation method was used. Data were cleaned and each variable was divided into relevant categories for age groups, socioeconomic status (SES), feeding practices, anthropometric measurements, and dental caries measurements. The Wealth index scores were broken down into quintiles to categorize participants by SES. Age data were based on months at baseline, and children were categorised into one of four age groups (<6 months, 6–12 months, 12–18 months, >18 months).

The Minimum Acceptable Diet (MAD) was calculated using the WHO Guidelines and is a composite index of breastfeeding or milk feedings, Minimum Diet Diversity and Minimum Meal Frequency; indicators being available for children aged 6 to 23.9 months at baseline. The Minimum Diet Diversity, after the 2010 guidelines, was based on whether or not the child consumed in the past 24 h a minimum 4 out of a list of 7 food groups. These food groups included: (1) Grains, roots and tubers; (2) Legumes and nuts; (3) Dairy products; (4) Flesh foods; (5) Eggs; (6) Vitamin A-rich fruits and vegetables; and (7) Other fruits and vegetables. The Minimum Meal Frequency was calculated on the minimum amount of times that the child received solid, semisolid or soft foods the previous day according to their age group and breastfeeding status [34,35].

As a Cambodian specific diet indicator, The Cambodian Complementary Feeding (CCF) variable was calculated based on the Nutrition Handbook for the family, prepared and adapted by the National Cambodian Nutrition Program [36]. It was based on the number of spoons of Chan Chang Koeh and determined according to age and breastfeeding status [29,30]. 

Anthropometric measurements were converted to sex-specific height/length-for-age *Z*-scores using the WHO Child Growth Standards 2016 [32], and identified as stunting at <−2SDs. The low proportion of obese children in the sample (<2%), identified at >+2SDs for weight-for-age, was insufficient in number to establish a comparison group. New cases of stunting were identified, i.e., children who had optimal HAZ at baseline, while at 1-year follow-up were identified with HAZ below −2 SDs. To assess caries experience, the Significant Caries Index (ScI) was calculated at baseline and follow-up by ranking individuals within age groups according to the number of teeth with carious lesions (based on the dmft index: decayed, missing or filled teeth). Subsequently, the mean of the most severe one-third of the population was calculated [37]. That value was then used to create an age-adjusted dichotomous variable to indicate the presence or absence of ScI. This index represents a more “severe” disease experience, as children with a high dmft score and presenting ScI would be more likely to develop lesions that might create mouth pain or infection, which are acknowledged consequences of carious lesions [12,13,14].

After cleaning and categorising the data, descriptive and multivariate analyses were performed. The Chi-squared test was used to examine differences in proportions among sociodemographic subgroups for stunting, caries, and diet indicators. Multivariate logistic regression was performed to explore the relationship between severe caries experience (by ScI) at baseline and follow-up with the onset of new cases of stunting over the observation period. Gender, province, age, SES, and the MAD and CCF variables for diet indicators were included in the multivariate modelling.

## 3. Results

### 3.1. Descriptive Statistics

#### 3.1.1. Demographic Profile and Clinical Characteristics at Baseline

Baseline participants’ characteristics are shown in Table 1. Three out of four participants came from the northeastern provinces of Kratie and Ratanakiri and one quarter from Phnom Penh. There was an even gender distribution. Two-thirds of participants were 1 to 2 years of age at baseline; and over one-third of participants belonged to the lowest two wealth index quintiles. Stunted growth was present in 25.6% of the children. Males, children from Ratanakiri province, those from the older age groups, and those from the second quintile, “low”, presented a significantly higher prevalence of stunting. Overall, 51.9% of the children presented with “any caries” at baseline and there was no statistically significant difference by gender. Those in Ratanakiri and in older age groups had a significantly higher prevalence of any caries and severe caries (by baseline ScI). When looking at SES, those in the medium quintile presented the lowest caries experience, and the lowest wealth index quintile had the higher proportion of children who had ScI at baseline compared to the other quintiles (21% vs. 12–14%, *p* = 0.040; χ^2^ test).

The mean (SD) dmft for children with caries at baseline was 5.1 (3.6). The mean cutoff of ScI by age groups at baseline was 4.7; with a mean dmft of 8.8 (4.1) for children with ScI and 1.4 (2.0) in children without. Children with stunting presented a dmft of 2.7 (3.5) and children without stunting 2.3 (3.6) (*p* = 0.053).

#### 3.1.2. Clinical Characteristics at Follow-Up

The participants’ characteristics at follow-up are shown in Table 2. Stunted growth was present in 39.9% of the children. In contrast to baseline, females had a significantly higher chance of presenting with stunting at follow-up (42.2% vs. 36.1%, *p* = 0.026 χ^2^ test). Children from Ratanakiri province, approximately >12 months of age, and from the “low” SES quintile, continued to present the highest prevalence of stunting. 

Overall, 63.6% of the children at follow-up presented dental caries. A higher proportion of males than females presented with ScI at follow-up (17.6 vs. 12.1%, *p* = 0.005; χ^2^ test). Children living in Phnom Penh experienced a greater increase in caries and demonstrated higher overall caries experience at follow-up compared to other provinces. Differences in caries experience were seen among the five SES groups, whereby the highest SES group had a greater proportion of children with any caries (75% vs. 49–66%) and higher mean dmft (*p* < 0.001; χ^2^ test, ANOVA). A higher prevalence of ScI among the lowest SES was no longer significant at follow-up.

The mean (SD) dmft for children with caries at follow-up was 5.1 (3.9). The mean cutoff of ScI by age groups at follow-up was 7.2, with a mean dmft of 10.6 (4.0) for children with ScI and 2.0 (2.2) in children without. The mean dmft in stunted children was 3.2 (3.9) and in non-stunted children 3.3 (4.1) (*p* = 0.800). 

#### 3.1.3. New Cases of Stunting, Significant Caries, and Dietary Adequacy by Sociodemographic Characteristics

Overall, 17.6% of children transitioned from normal values at baseline to suboptimal level of length/height-for-age at follow-up. Children who presented stunted malnutrition at baseline were excluded from the analysis (*n* = 332). Females were significantly more likely to transition to a suboptimal level of height-for-age at follow-up; one out of four females and one out of five males presented new cases of stunting (*p* = 0.001; χ^2^ test). Children in Ratanakiri had a higher proportion of new cases of stunting when compared to other provinces (*p*-value ≤ 0.001; χ^2^ test). Children from the youngest age group, at approximately 18 months at follow-up, presented the highest frequency of new cases of stunting (*p*-value < 0.001, χ^2^ test). Regarding SES, the children from the second wealth index quintile (low) presented the highest prevalence of new cases of stunted malnutrition (*p*-value ≤ 0.001; χ^2^ test).

Measures of dietary adequacy by sociodemographic characteristics are presented in Table 3. Males were more likely than females to have adequate dietary intake, including a significantly higher likelihood of achieving CCF (62.6% vs. 51.3%, *p* < 0.001; χ^2^ test). Phnom Penh had the highest proportion, and Ratanakiri the lowest proportion of children who met the criteria for MAD and CCF. Children 0–6 months of age presented the highest proportion of acceptable CCF intake (*p* < 0.001; χ^2^ test). Children 12 months and over were more likely than children age 6–12 months to have received MAD (*p* < 0.001; χ^2^ test). Children in the first 4 SES quintiles demonstrated a progressively higher chance of meeting dietary criteria by MAD and CCF; however, those in the highest quintile appeared to have a lower dietary quality than those in the high quintile (*p* < 0.001; χ^2^ test).

### 3.2. Multivariate Analysis: Logistic Regression

Results of multivariate logistic regression on odds ratios for new cases of stunting based on severe dental caries by ScI at baseline and follow-up are presented in Table 4 (Models 1 and 2).

Having severe caries by ScI at baseline was not associated with developing new-onset stunting, although children with ScI at follow-up had approximately twice the risk with an odds ratio of 1.8 (95% CI = 1.0–3.0, *p* = 0.039; χ^2^ test) after controlling for gender, province, age, SES, and diet indicators. The highest risk for developing new cases of stunting was seen in females (OR 1.7; CI 1.2, 2.6) from rural areas (OR 2.5–3.3; CI 1.4, 6.0), at the age of approximately 1–2 years at follow-up (<12 months at baseline) (OR 0.4–0.5; CI 0.3, 0.9), and coming from the second lowest SES quintile (OR 2.0; CI 1.0, 3.8), even in the presence of MAD (OR 1.9; CI 1.1, 3.1).

Collinearity between MAD and CCF is taken into consideration and ruled out because both variables capture different aspects of the feeding practices.

## 4. Discussion

This secondary analysis of longitudinal data on Cambodian children under age 2 at baseline and 1 year later at follow-up showed that children with severe dental caries (as indicated by the ScI) had almost twice the odds of developing chronic malnutrition. This suggests that dental caries experience could be an important contributor to child growth restriction at a critical stage of child development. The results of the present analysis align with some cross-sectional and longitudinal studies that reported a positive association between severe dental caries and different forms of undernutrition [38,39,40,41]. The present study is unique in that it uses a longitudinal study design, controlled for the children’s dietary intake, and included children under 2 years of age at baseline from a population with severe caries experience.

A number of systematic reviews have examined the relationship between dental caries and malnutrition; some studies show association of caries with undernutrition, others with obesity, and others show no association [42,43,44,45,46,47]. The differences in child age, country’s socioeconomic standing, and socioeconomic and behavioural disparities within the countries may explain the disparate findings in the literature. Moreover, the caries–malnutrition relationship appears to play out differently in urban and rural areas, different SES groups, and different countries and regions based on their stage in the nutrition transition [48,49,50,51,52]. The data from this study suggested that children in predominantly rural northeastern provinces had worse baseline dietary, anthropometric, and caries experience. However, children in the predominantly urban Phnom Penh presented with worse caries experience over the observation period, which may be explained by higher sugar consumption.

Suboptimal dietary, anthropometric, and caries indicators were seen in the low/lowest SES population; however, the highest SES population did not present the best outcomes—the wealthiest SES group demonstrated lower rates of dietary adequacy and higher rates of dental caries and undernutrition compared to the adjacent lower SES strata. Furthermore, some children in the higher SES strata may be vulnerable to overnutrition [53], but the limited number of obese participants in the sample precluded analysis of the relationship of severe dental caries and overnutrition. There is consensus that frequent consumption of high-calorie and sugary foods and drinks contributes to dental caries and overweight [6,54]. On the other hand, consuming sugary foods and drinks may also displace the intake of nutritious foods and drinks, and contribute to dental caries and undernutrition [55,56]. This suggests that, in low-middle-income countries, there may be a “u-shaped” association between SES and dietary adequacy, caries and nutritional status, rather than the linear relationship between SES and good nutrition and oral health generally described in higher-income countries [53,57,58]. Nonetheless, there is a paucity of research about the role of sugary diets in undernutrition, including the comorbidity with dental caries. This may be explained by the lack of oral health awareness among non-dental medical and public health professionals [59], and the economic power of the food and beverage companies that sell sugary snacks and beverages as well as oral health care products, and sponsor caries research that focuses on dental care products rather than on the cause of caries: sugar [60].

In addition, a growing body of research highlights gender inequalities in child nutrition and health [61,62]. In this sample, females showed worse diet and anthropometric outcomes, and males presented more severe caries experience at follow-up. However, Cambodian national data show no substantial difference in nutritional outcomes by gender in children under 5 [18,63]. The findings also support the understanding that both dental caries and chronic undernutrition are socially driven phenomena with complex and poorly studied interrelationships with each other and with sociocultural, economic and behavioural factors. In this study sample, it appears that male infants/toddlers, and those in the higher SES strata may have consumed larger quantities of food as implied by the CCF data, which may protect first against growth faltering; however, the same subgroups may also have greater consumption of sugary foods and drinks, thereby contributing to the finding of more severe caries experience, and associated long-term growth faltering. This highlights the need to address the structural drivers of early childhood caries and undernutrition from early infancy onward to reduce the long-term adverse impacts of dental caries on children’s nutrition, health, and development [13].

Several limitations should be recognized in this study. The CAHENMS and the oral health survey were not specifically intended to examine the impact of tooth decay on malnutrition, and lacked detailed data on maternal health, education, and comorbidities that might contribute to the higher risk for stunting in the presence of severe caries. The dietary surveys were subject to recall bias and the diet indicators of this secondary analysis presented some constraints. The WHO indicator, Minimal Acceptable Diet (MAD) is not country-specific, and many Cambodian food items were missed. Moreover, the Minimal Diet Diversity (MDD) does not include breast milk as it has been recently proposed in the most recent guidelines. The Cambodian Complementary Feeding (CCF) indicator includes the quantity of complementary feeding and breastfeeding but not the frequency of meals per day. Thereby, the diet indicators may not have fully captured children’s diet in Cambodia and data were not collected on daily child consumption of non-nutritious food and drinks (junk food). Those food groups contain high amounts of sugar, and are hypothesized to be primary causes of caries and associated stunted growth. Finally, as malnutrition and dental caries are multifactorial diseases that evolve over time [64], this one-year follow-up does not capture the cumulative impact of disease, the mitigating effect of oral hygiene and dental treatment, and a longer follow-up considering these factors is warranted. This would also allow for change modelling of trends in growth and development for which binary logistic regression may not be sensitive.

In order to use human and financial resources wisely, it is important to work closely with programmes focused on the prevention of non-communicable diseases (NCDs) that share common risk factors with tooth decay [65]. Although many countries have introduced school-based caries-prevention programmes with successful outcomes [66,67], most caries risk factors and adverse outcomes begin in early childhood, before the child attends school and before the child sees a dentist. Thus, there is an evidence-based opportunity to integrate caries prevention into existing primary care maternal-child health programmes from birth onwards [68]. 

The findings of this study suggest that reducing the severity of dental caries could, in turn, reduce the long-term adverse impact on linear growth among young children. Therefore, it could be justified to incorporate caries prevention initiatives with general child nutrition and health promotion initiatives. Such interventions might include supporting adequate infant breast- and complementary feeding [69,70], babyWASH and consumption of safe water [71], and healthy environments with restricted access to sugar [72]. This should go along with an improvement of oral hygiene practices—toothbrushing two times a day with fluoride toothpaste [73]—and oral health initiatives such as population-based fluoride [74,75], and preventing caries lesions through application of sealants [76], fluoride varnishes [73,77], and topical silver and fluoride agents [78]. Finally, particularly relevant to the Cambodian context of widespread food insecurity is the need for political will to limit the global marketing of low-cost, ultra-processed junk food and sugar-sweetened drinks to improve child nutritional status, decrease the incidence of dental caries, and ultimately support children’s growth and development.

## 5. Conclusions

This is the first longitudinal study to examine the association between dental caries and stunted growth in children under 2 years in Cambodia, and to demonstrate that severe tooth decay is associated with developing stunting malnutrition. The study highlights the need to prevent and treat early childhood tooth decay as an important part of programmes to prevent child undernutrition as well as NCDs, and to promote children’s optimal growth and development during a critical stage of life.

## Figures and Tables

**Figure 1 nutrients-13-00290-f001:**
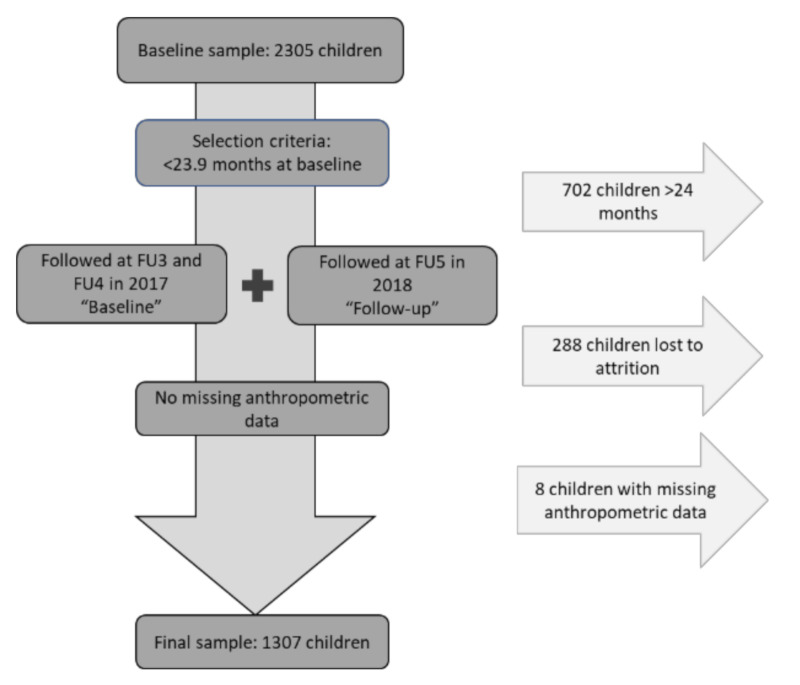
Flow chart depicting selection criteria of individuals included in the final sample size.

**Table 1 nutrients-13-00290-t001:** Sociodemographic and clinical characteristics at baseline.

	Total	Stunting Baseline		Any Caries Baseline		ScI ^1^ Baseline	
	*N* (Row %)	*N* (Row %)		*N* (Row %)		*N* (Row %)	
		Yes	No	Yes	No	Yes	No
Total	1307 (100)	332 (25.4)	975 (74.6)	629 (51.9)	678 (48.1)	188 (14.4)	1119 (85.6)
Gender							
Male	631 (48.3)	179 (28.4)	452 (71.6)	316 (50.1)	315 (49.9)	100 (15.8)	531 (84.2)
Female	676 (51.7)	153 (22.6)	523 (77.4)	313 (46.3)	363 (53.7)	88 (13.0)	588 (87.0)
*p*-value ^2^			0.017		0.172		0.145
Province							
Phnom Penh	319 (24.4)	39 (12.2)	280 (87.8)	153 (48.0)	166 (52.0)	39 (12.2)	280 (87.8)
Kratie	551 (42.2)	136 (24.7)	415 (75.3)	235 (42.6)	316 (57.4)	66 (12.0)	485 (88.0)
Ratanakiri	437 (33.4)	157 (35.9)	280 (64.1)	241 (55.1)	196 (44.9)	83 (19.0)	354 (81.0)
*p*-value ^2^			<0.001		<0.001		0.003
Age at baseline							
<6 months	143 (10.9)	17 (11.9)	126 (88.1)	15 (10.5)	128 (89.5)	15 (10.5)	128 (89.5)
6–12 months	278 (21.3)	53 (19.1)	225 (80.9)	71 (25.5)	207 (74.5)	45 (16.2)	233 (83.8)
12–18 months	438 (33.5)	116 (26.5)	322 (73.5)	246 (56.2)	192 (43.8)	65 (14.8)	373 (85.2)
18–24 months	448 (34.3)	146 (32.6)	302 (67.4)	297 (66.3)	151 (33.7)	63 (14.1)	385 (85.9)
*p*-value ^2^			<0.001		<0.001		0.456
SES							
Lowest	220 (16.8)	71 (32.3)	149 (67.7)	118 (53.6)	102 (46.4)	46 (20.9)	174 (79.1)
Low	265 (20.3)	83 (31.3)	182 (68.7)	135 (50.9)	130 (49.1)	34 (12.8)	231 (87.2)
Medium	393 (30.1)	91 (23.2)	302 (76.8)	171 (43.5)	222 (56.5)	56 (14.2)	337 (85.8)
High	215 (16.4)	34 (15.8)	181 (84.2)	115 (53.5)	100 (46.5)	25 (11.6)	190 (88.4)
Highest	214 (16.4)	53 (24.8)	161 (75.2)	90 (42.1)	124 (57.9)	27 (12.6)	187 (87.4)
*p*-value ^2^			<0.001		0.014		0.040
Caries							
Dmft mean (SD)		2.7 (3.5)	2.3 (3.6)	5.1 (3.6)		8.8 (4.1)	1.4 (2.0)
*p*-value ^2^			0.053				0.001

^1^ ScI: Significant Caries Index. ^2^ χ^2^ test for differences among groups within the same columns or ANOVA. SES: socioeconomic status.

**Table 2 nutrients-13-00290-t002:** Sociodemographic and clinical characteristics at follow-up.

	Stunting Follow-Up		Any Caries Follow-Up		Sci ^1^ Follow-Up	
	*N* (Row %)		*N* (Row %)		*N* (Row %)	
	Yes	No	Yes	No	Yes	No
Total	513 (39.3)	794 (60.7)	831 (63.6)	476 (36.4)	193 (14.8)	1114 (85.2)
Gender						
Male (631)	228 (36.1)	403 (63.9)	401 (63.5)	230 (36.5)	111 (17.6)	520 (82.4)
Female (676)	285 (42.2)	391 (57.8)	430 (63.6)	246 (36.4)	82 (12.1)	594 (87.9)
*p*-value ^3^		0.026		0.982		0.005
Province						
Phnom Penh (319)	74 (23.2)	245 (76.8)	255 (79.9)	64 (20.1)	60 (18.8)	259 (81.2)
Kratie (551)	206 (37.4)	345 (62.6)	318 (57.7)	233 (42.3)	79 (14.3)	472 (85.7)
Ratanakiri (437)	233 (53.3)	204 (46.7)	258 (59.9)	179 (41.0)	54 (12.4)	383 (87.6)
*p*-value ^3^		<0.001		<0.001		0.044
Approx. age follow-up ^2^						
<12 months (143)	74 (51.7)	69 (48.3)	52 (36.4)	91 (63.6)	32 (22.4)	111 (77.6)
12–18 months (278)	109 (39.2)	169 (60.8)	148 (53.2)	130 (46.8)	15 (5.4)	263 (94.6)
18–24 months (438)	160 (36.5)	278 (63.5)	283 (64.6)	155 (35.4)	68 (15.5)	370 (84.5)
24–36 months (448)	170 (37.9)	278 (62.1)	348 (77.7)	100 (22.3)	78 (17.4)	370 (82.6)
*p*-value ^3^		0.011		<0.001		<0.001
SES						
Lowest (220)	99 (45.0)	121 (55.0)	107 (48.6)	113 (51.4)	21 (9.5)	199 (90.5)
Low (265)	147 (55.5)	118 (44.5)	168 (63.4)	97 (36.6)	37 (14.0)	228 (86.0)
Medium (393)	130 (33.1)	263 (66.9)	259 (65.9)	134 (34.1)	67 (17.0)	326 (83.0)
High (215)	58 (27.0)	157 (73.0)	136 (63.3)	79 (36.7)	34 (15.8)	181 (84.2)
Highest (214)	79 (36.9)	135 (63.1)	161 (75.2)	53 (24.8)	34 (15.9)	180 (84.1)
*p*-value ^3^		<0.001		<0.001		0.140
Caries						
Dmft mean (SD)	3.2 (3.9)	3.3 (4.1)	5.1 (3.9)		10.6 (4.0)	2.0 (2.2)
*p*-value ^3^		0.800				<0.001

^1^ ScI: Significant Caries Index. ^2^ Approximate age mean (SD) at follow-up: 30.79 (6.52). ^3^ χ^2^ test for differences among groups within the same columns or ANOVA.

**Table 3 nutrients-13-00290-t003:** Sociodemographic characteristics of new cases of stunting, severe caries and dietary adequacy.

	New Cases Stunting		ScI Baseline		ScI Follow-Up		CCF ^1^		MAD ^2^	
	Valid: 975 Missing: 332 *N* (Row %)		Valid: 1307 Missing: 0 *N* (Row %)		Valid: 1307 Missing: 0 *N* (Row %)		Valid: 1127 Missing: 180 *N* (Row %)		Valid: 1017 Missing: 290 *N* (Row %)	
	Yes	No	Yes	No	Yes	No	Yes	No	Yes	No
Total	230 (17.6)	745 (57.0)	188 (14.4)	1119 (85.6)	193 (14.8)	1114 (85.2)	201 (15.4)	816 (62.4)	640 (49.0)	487 (37.3)
Gender										
Male (631)	85 (18.8)	367 (81.2)	100(15.8)	531 (84.2)	111 (17.6)	520 (82.4)	113 (22.1)	398 (77.9)	343 (62.6)	205 (37.4)
Female (676)	145 (27.7)	378 (72.3)	88 (13.0)	588 (87.0)	82 (12.1)	594 (87.9)	88 (17.4)	418 (82.6)	297 (51.3)	282 (48.7)
*p*-value ^3^		0.001		0.145		0.005		0.059		<0.001
Province										
Phnom Penh (319)	42 (15.0)	238 (85.0)	39 (12.2)	280 (87.8)	60 (18.8)	259 (81.2)	94 (38.5)	150 (61.5)	201 (71.3)	81 (28.7)
Kratie (551)	97 (23.4)	317 (76.6)	66 (12.0)	485 (88.0)	79 (14.3)	472 (85.7)	77 (17.9)	357 (82.1)	257 (53.3)	225 (46.7)
Ratanakiri (437)	91 (32.5)	189 (67.5)	83 (19.0)	354 (81.0)	54 (12.4)	383 (87.6)	30 (8.7)	313 (91.3)	182 (50.1)	181 (49.9)
*p*-value ^3^		<0.001		0.003		0.044		<0.001		<0.001
Approx. age follow-up										
<12 months (143)	58 (46.0)	68 (54.0)	15 (10.5)	128 (89.5)	32 (22.4)	111 (77.6)	/	/	106 (89.1)	13 (10.9)
12–18 months (278)	63 (28.0)	162 (72.0)	45 (16.2)	233 (83.8)	15 (5.4)	263 (94.6)	28 (10.9)	228 (89.1)	122 (67.8)	58 (32.2)
18–24 months (438)	55 (17.1)	267 (82.9)	65 (14.8)	373 (85.2)	68 (15.5)	370 (84.5)	92 (23.3)	303 (76.7)	167 (40.0)	251 (60.0)
24–36 months (448)	54 (17.9)	248 (82.1)	63 (14.1)	385 (85.9)	78 (17.4)	370 (82.6)	81 (22.1)	225 (77.9)	245 (59.8)	165 (40.2)
*p*-value ^3^		<0.001		0.456		<0.001		<0.001		<0.001
SES										
Lowest (220)	35 (23.5)	114 (76.5)	46 (20.9)	174 (79.1)	21 (9.5)	199 (90.5)	23 (12.3)	164 (87.7)	79 (43.6)	102 (56.4)
Low (265)	77 (42.3)	105 (57.5)	34 (12.8)	231 (7.2)	37 (14.0)	228 (86.0)	22 (11.2)	175 (88.8)	139 (61.2)	88 (38.8)
Medium (393)	57 (18.9)	245 (81.1)	56 (14.2)	337 (85.8)	67 (17.0)	326 (83.0)	57 (20.0)	228 (80.0)	196 (59.9)	131 (40.1)
High (215)	28 (15.5)	153 (84.5)	25 (11.6)	190 (88.4)	34 (15.8)	181 (84.2)	60 (34.9)	112 (65.1)	139 (69.5)	61 (30.5)
Highest (214)	33 (20.5)	128 (79.5)	27 (12.6)	187 (87.4)	34 (15.9)	180 (84.1)	39 (22.2)	137 (77.8)	87 (45.3)	105 (54.7)
*p*-value ^3^		<0.001		0.040		0.140		<0.001		<0.001

^1^ CCF: Cambodian complementary feeding. ^2^ MAD: Minimum acceptable diet. ^3^ χ^2^ test for differences among groups within the same columns or ANOVA.

**Table 4 nutrients-13-00290-t004:** Logistic regression models for the risk of stunting by caries experience, sociodemographic and diet indicators.

	New Cases Stunting				
	Based on ScI at Baseline MODEL 1			Based on ScI at Follow-Up MODEL 2	
	OR (95% CI)	*p*-Value		OR (95% CI)	*p*-Value
Caries experience					
No ScI ^1^					
Had ScI	0.8 (0.5–1.5)	0.524		1.8 (1.0–3.0)	0.039
Gender					
Male ^1^					
Female	1.7 (1.1–2.5)	0.012		1.7 (1.2–2.6)	0.009
Province					
Phnom Penh ^1^					
Kratie	2.3 (1.3–4.0)	0.003		2.5 (1.4–4.3)	0.002
Ratanakiri	3.1 (1.7–5.6)	<0.001		3.3 (1.8–6.0)	<0.001
Age at baseline					
6–12 months ^1^					
12–18 months	0.5 (0.3–0.8)	0.009		0.4 (0.3–0.8)	0.004
18–24 months	0.6 (0.4–1.0)	0.068		0.5 (0.3–0.9)	0.027
SES					
Lowest ^1^					
Low	2.0 (1.0–3.7)	0.040		2.0 (1.0–3.8)	0.038
Medium	0.7 (0.3–1.3)	0.211		0.7 (0.3–1.2)	0.195
High	1.0 (0.5–2.0)	0.964		1.0 (0.5–2.0)	0.978
Highest	1.3 (0.6–2.5)	0.512		1.3 (0.7–2.6)	0.422
Diet indicators					
No CCF ^1^					
CCF	1.2 (0.8–1.9)	0.333		1.2 (0.8–1.9)	0.367
No MAD ^1^					
MAD	1.9 (1.2–3.2)	0.011		1.9 (1.1–3.1)	0.014

^1^ Reference value.

## Data Availability

Restrictions apply to the availability of these data. Data was obtained from the Health and Nutrition Monitoring Study (CAHENMS) in the North East of Cambodia 2016-2019 and are available with the permission of the principal investigator.

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
