# Peer review of "Stunting Malnutrition Associated with Severe Tooth Decay in Cambodian Toddlers"

_nutrients, 2021, doi:10.3390/nu13020290_

Round 1

Reviewer 1 Report

The manuscript seems to be interesting but I have some comments.

Title- ...severe tooth decay... How did you assessed severe tooth decay?

dmft index does not distinguish dental caries between severe and not severe. Dmft tells about the dental caries only.

What is severe dental caries and how this was assessed because it can not be assessed by dmft.

This index does not assess chronic inflamation, chronic oral infection (line 49), dental sepsis (line 59). How this was assessed because authors often use this phrases?

Why there is nothing about ECC (early childhood caries). All this participants suffered from ECC.

Why there was no information about breastfeeding which has positive influence regarding ECC?

How mouth pain was assessed and chronic inflamation, not by dmft?

Dental caries is NCD and behavioral disease and does not cause dissolution tooth enamel (line 45). Erosion causes tooth dissolution but erosion is not dental caries. This two diseases have different onset !

Dental caries is a multifactorial disease. The most important is biofilm- there is nothing about the role of biofilm in the manuscript. Removal of biofilm is essential in prevention of dental caries together with delivery of fluoride (there is nothing about this) and limited access to carbohydrates and normal saliva secretion.

There was something about SiC, it would be interesting to know more about this.

Diet- there is only information about diet questionnary, but I think this should be in detali, because word diet does not tell much to the reader and everybody can think about it in a different way.

In Discussion line 69-84- there is information about prevention measures but nothing about tooth brushing which is considered as a primary prevention against dental caries

I think other causes of stunting malnutrition should be taken under consideration as well and discussed in the manuscript

Reviewer 2 Report

The article is appropriately designed, well-written and presented. The study calculated the Minimum Acceptable Diet (MAD) registering diet diversity and meal frequency in children. However, did not mention the daily intake of total sugar from sugary soft drinks and junk foods. It would have been better if the authors have estimated the intake of total amount of sugar to construct a strong association between the sugar intake, dental caries, stunting and malnutrition in children. The authors calculated “The Cambodian Complementary Feeding (CCF) variable” based on the number of spoons of Chan Chang Koeh, a traditional bowl in Southeast Asia, that could be used to calculate the overall sugar and nutrient intake.

The study concludes the need to prevent and treat early childhood tooth decay as an important part of programs to prevent child under-nutrition as well as NCDs but has not given any suggestions and recommendations the dietary and oral hygiene practices. A few typos/editorial corrections are required and should be taken care. Some of sentences do not express the meaning correctly and must be looked for clarity in language.

Round 2

Reviewer 1 Report

Point 1 SiC is 30% of the highest dmft. Please add what is the value of SiC in the participants and what kind of prevention measures can be apply to reduce it.

Point 2 This is private opinion please add a reference. Point 5 not agree, there is an index that assesses pain and inflamation called pufa. You can not draw far feched conclusions. Point 8 can you add what are the recommendations in your country regarding toothbrushing.This would be interesting.
